# Loss of Mitochondrial Tusc2/Fus1 Triggers a Brain Pro-Inflammatory Microenvironment and Early Spatial Memory Impairment

**DOI:** 10.3390/ijms25137406

**Published:** 2024-07-05

**Authors:** Tonie Farris, Salvador González-Ochoa, Muna Mohammed, Harshana Rajakaruna, Jane Tonello, Thanigaivelan Kanagasabai, Olga Korolkova, Akiko Shimamoto, Alla Ivanova, Anil Shanker

**Affiliations:** 1Department of Biomedical Sciences, School of Graduate Studies, Meharry Medical College, Nashville, TN 37208, USA; tfarris20@email.mmc.edu (T.F.); mmohammed22@email.mmc.edu (M.M.); tkanagasabai@mmc.edu (T.K.); 2Department of Biochemistry, Cancer Biology, Neuroscience & Pharmacology, School of Medicine, Meharry Medical College, Nashville, TN 37208, USA; sgonzalezochoa@mmc.edu (S.G.-O.); jtonello@mmc.edu (J.T.); okorolkova@mmc.edu (O.K.); medfordma02155@gmail.com (A.S.); 3The Office for Research and Innovation, Meharry Medical College, Nashville, TN 37208, USA; hrajakaruna@mmc.edu

**Keywords:** aging, neuroinflammation, cognitive impairment, Tusc2, Fus1, mitochondria, calcium, brain immune populations, sex-dependent changes

## Abstract

Brain pathological changes impair cognition early in disease etiology. There is an urgent need to understand aging-linked mechanisms of early memory loss to develop therapeutic strategies and prevent the development of cognitive impairment. Tusc2 is a mitochondrial-resident protein regulating Ca^2+^ fluxes to and from mitochondria impacting overall health. We previously reported that *Tusc2*^−/−^ female mice develop chronic inflammation and age prematurely, causing age- and sex-dependent spatial memory deficits at 5 months old. Therefore, we investigated Tusc2-dependent mechanisms of memory impairment in 4-month-old mice, comparing changes in resident and brain-infiltrating immune cells. Interestingly, *Tusc2*^−/−^ female mice demonstrated a pro-inflammatory increase in astrocytes, expression of IFN-γ in CD4^+^ T cells and Granzyme-B in CD8^+^T cells. We also found fewer FOXP3^+^ T-regulatory cells and Ly49G^+^ NK and Ly49G^+^ NKT cells in female Tusc2^−/−^ brains, suggesting a dampened anti-inflammatory response. Moreover, *Tusc2*^−/−^ hippocampi exhibited Tusc2- and sex-specific protein changes associated with brain plasticity, including mTOR activation, and Calbindin and CamKII dysregulation affecting intracellular Ca^2+^ dynamics. Overall, the data suggest that dysregulation of Ca^2+^-dependent processes and a heightened pro-inflammatory brain microenvironment in *Tusc2*^−/−^ mice could underlie cognitive impairment. Thus, strategies to modulate the mitochondrial Tusc2- and Ca^2+^- signaling pathways in the brain should be explored to improve cognitive health.

## 1. Introduction

Alzheimer’s disease (AD) is a complex, multifactorial disease affecting many individuals worldwide. The major risk factor for AD development is aging [1]. According to 2023 data from the World Health Organization (WHO), more than 55 million people have dementia worldwide, and every year, nearly 10 million new cases are added. Women are disproportionately affected by dementia [2]. Women are reported to have a higher incidence of MCI (Mild Cognitive Impairment) and AD than men [3]. Longer life expectancy as well as the protective role of hormone estradiol diminishing with age are the factors that are suspected to play a role in higher disease incidence in females than males [4,5,6]. However, the precise mechanisms of aging-linked AD and female susceptibility to AD are still not established but urgently needed.

The first clinical stage of AD, MCI, is characterized by a decline in cognitive function that mostly does not interfere with activities of daily living. Notably, only 30% of people with MCI progress to an irreversible stage of the disease characterized by severe memory loss and loss of independence [7]. Thus, understanding the mechanisms of MCI that drive the disease to an irreversible stage and developing early diagnostic tools and approaches to treat the primary cause of AD early, before amyloid plaque formation, is of the utmost importance.

Tusc2 (alternative name is Fus1) is an evolutionarily conserved, mitochondrial, nuclear-encoded protein expressed in every cell. We identified its function in mitochondria as a regulator of mitochondrial Ca^2+^ fluxes and, thus, energy and overall mitochondrial health [8,9]. Tusc2 function is especially critical in cells with high energy demands, such as immune and neural cells [8,10]. We established that the loss of *Tusc2* in mice results in chronic inflammation [11], dysregulation of immune responses [8,9,12], and defects in NK cell maturation [13]. In later studies, we characterized Tusc2 mice as a model of premature aging, early olfactory and spatial memory impairments, and progressive age-dependent hearing impairment linked to mitochondrial deficiency. The majority of these signs are consistent with aging-linked MCI symptoms in humans and are the hallmarks of many neurodegenerative diseases [14]. Importantly, we established that a 50% reduction in Tusc2 activity is sufficient to induce disease [13]. Our published and preliminary data linked a decrease in Tusc2 in immune cells with inflammation [8]. Taken together, we hypothesize that *Tusc2* KO mice could be a valuable model for characterizing early mechanisms and processes involved in the development of MCI and progression from MCI to dementia [15]. 

In this study, we address the Tusc2-dependent brain immune changes that may underlie cognitive impairment (Figure 1), focusing on the early stages of the disease in 4-month-old males and females. Currently, the roles of brain-resident and brain-infiltrating immune cells in memory loss are an area of active research. Immune cells such as T cells, B cells, NK cells, and monocytes can infiltrate the brain in response to signals of inflammation or tissue damage [16]. Infiltrating immune cells can contribute to the neuroinflammatory response initiated by resident microglia and astrocytes and release pro-inflammatory cytokines, modulating or amplifying the inflammatory cascade in the brain [17]. Thus, the brain’s immune cells may play protective and pathological roles in developing the disease.

Most importantly, we characterize Tusc2-, age-, and sex-dependent immune changes in the brain of *Tusc2* KO mice that may provide new knowledge on the cause of early memory impairment. Finally, we characterize important, AD-specific molecular changes in the hippocampus, a part of the brain involved in short-term memory consolidation and storage. These studies will be instrumental in understanding the immune mechanisms of memory impairment and their differences between males and females.

## 2. Results

### 2.1. Loss of Tusc2 Caused Deficits in Short-Term Spatial Memory

To compare the effects of *Tusc2* loss in male and female mice on cognitive functions, 4-month-old KO mice of both sexes were subjected to a battery of behavioral tests (Figure 2A). Tusc2 KO males showed significant short-term spatial memory deficits in Y-maze tests based on a lower percentage of correct alternations as compared to their WT male counterparts (F_(1, 52)_ = 12.74, *p* = 0.0055) (Figure 2B). *Tusc2* KO 4-month-old females did not show a significant difference in Y-maze test performance as compared to WT mice (*p* = 0.3866). The total number of entries during testing was also recorded, and no a significant difference was found across groups (Figure 2C). Recognition memory was evaluated using the novel object recognition (NOR) test (Figure 2D). We found no Tusc2- or sex-dependent differences in the performance of this test. Finally, we used the open-field test to evaluate locomotor function and anxiety-like behavior; however, the time spent in the peripheral and central zone showed no significant difference between Tusc2 KO and WT mice of both sexes (Figure 2E,F), suggesting no anxiety or locomotor function deficiency in Tusc2 KO mice of both sex. 

Thus, we demonstrated in this study that 4-month-old *Tusc2* KO, but not WT males, have spatial memory deficits. In our previous study, we demonstrated that 5-month-old *Tusc2* KO, but not WT females, have significant memory deficits [18]. Since inflammation, particularly chronic inflammation, is a critical pathology linked to memory impairment [19], we set out to investigate neuroinflammation in the *Tusc2* KO mice of both sexes via analysis of brain immune cells. 

### 2.2. Tusc2- and Sex-Dependent Changes in Brain Immune Populations

To investigate the impact of sex and *Tusc2* deficiency on brain health and the central nervous system (CNS), we evaluated key immune populations associated with neurological function and CNS defense mechanisms [20,21,22,23,24,25,26,27,28,29,30,31,32,33,34,35,36,37,38,39,40,41,42,43,44,45,46] (Appendix A). We analyzed brain-resident cells, including microglia and astrocytes, which function as the primary immune effector cells of the CNS and reside mainly in the brain parenchyma [47,48]. We also analyzed the populations of innate and adaptive immune cells that presumably come from the periphery in response to pro-inflammatory changes in the brain [49] (Appendix A). Flow data are expressed as the overall percentage or proportion of cells (%) and Mean Fluorescence Intensity (MFI) of cytokine expression. The percentage indicates the population’s proportion, while MFI shows the relative number of molecules produced or expressed by a cell [50]. It is worth noting that although both parameters are determined using the same detection channel, they characterize different parameters.

#### 2.2.1. *Tusc2* Deficiency Causes Astrogliosis and Increased Pro-Inflammatory Immune Subtypes

Microglia support brain immune homeostasis, acting as critical immune effector cells in the CNS [51]. At the same time, astrocytes participate in neurotransmitter uptake and recycling, inflammation, synaptic activity, and BBB maintenance [48]. Thus, we evaluated the proportion of microglia (CD11b^+^/CD45^+^/TMEM119^+^/F4/80^+^) and activated microglia based on the expression levels of CD45^+^ in the total microglial population in the brains of *Tusc2* WT and KO male and female mice (Figure 3A); a significant difference was observed in activated microglial populations only between WT mice (*p* = 0.0356) (Figure 3B). However, upon analyzing the proportions of activated (CD45^bright^/CD11b^+^) and resting microglia (CD45^dim^/CD11b^+^) [52,53] (Figure 3C), we found that the percentage of the activated subpopulation in female Tusc2 KO mice was significantly higher as compared to the female WT (*p* < 0.0001) and male Tusc2 KO mice (*p* = 0.0006) (Figure 3D). No significant changes were observed in the percentage of resting microglia between the groups (Figure 3E). These results demonstrate that even when the proportion of microglia is equal between the WT and *Tusc2* KO males and females, the female *Tusc2* KO brain microenvironment contains some factors that cause microglial activation. Moreover, analyzing the proportion of astrocytes (CD11b^+^/CD45^+^/ACSA2^+^) (Figure 3F), we revealed a ~2-fold increase in the percentage of astrocytes in females Tusc2 KO (*p* = 0.0002) vs. WT (Figure 3G).

The proportion of cytotoxic CD8 T, CD4 T helper, Treg, NK, and NKT cells was also evaluated. The number of activated CD4 T cells (CD4^+^/CD25^+^) in *Tusc2* KO (Figure 4A) shows that males were similar to their WT counterparts. In contrast, in *Tusc2* KO females, there was a significant decrease in the number of CD4^+^/CD25^+^ T cells between groups (*p* < 0.0001) and between the WT sex (*p* = 0.0227) (Figure 4B). 

Th_1_ cells are pro-inflammatory T cells that have been observed in the brains of Parkinson’s disease patients and experimental animal models of EAE [54,55]. To identify the T helper cell subset (Th_1_), we analyzed IFN-γ expression in CD4^+^CD3^+^ T cells (Figure 4C). The overall Th_1_ cell population size was not significantly different between the groups (Figure 4D). However, MFI revealed a significant increase in the expression levels of IFN-γ in Th_1_ cells from female *Tusc2* KO brains (*p =* 0.0068) (Figure 4E). No significant difference was seen in KO males. 

CD8^+^ T cells are considered pro-inflammatory and cytotoxic, although some early evidence reported their immune-suppressive function during neurodegeneration [56]. By gating the cell population (CD8^+^/CD25^+^) (Figure 5A), we showed that the CD8^+^ T population in the brain of *Tusc2* KO, of both males and females, have significantly decreased (*p =* 0.0006 and *p* < 0.0001, respectively) compared to their WT counterparts (Figure 5B). We also analyzed the effector subtypes of CD8^+^ T cells (CD3^+^/CD8^+^/CD28^+^/Granzyme B^+^) (Figure 5C). Interestingly, we observed a significant decrease in the number of this population in both male (*p* < 0.0001) and female *Tusc2* KO mice (*p =* 0.0003) as compared to their WT counterparts. We also observed significant differences between the sexes of *Tusc2* KO mice (*p =* 0.0032) (Figure 5D–F). However, we detected a trend toward increased expression (MFI) of the granzyme B in CD3^+^/CD8^+^ T cells in Tusc2 KO females (Figure 5G). These findings suggest that the *Tusc2* deficiency causes chronic pro-inflammatory processes in the brain through the action of both resident and infiltrating immune cells, which may consequently lead to MCI and neurodegeneration.

#### 2.2.2. *Tusc2* Deficiency Impairs Anti-Inflammatory Immune Populations

Guided by the pro-inflammatory changes we identified in the brain of *Tusc2* KO mice, we assessed immune inhibitory profiles, the other branch of the inflammation process. Treg cells dampen the inflammatory responses of effector T cells under normal physiological conditions. The populations of Treg cells (FoxP3^+^/CD3^+^/CD4^+^) were evaluated in male and female WT and *Tusc2* KO mice (Figure 6A). Flow cytometry analysis revealed that the proportion of Tregs was significantly decreased in male (*p =* 0.0001) and female (*p* < 0.0001) *Tusc2* KO mice as compared to their WT counterparts. We also observed significant differences between the sexes of *Tusc2* KO mice (*p* < 0.0001) (Figure 6B). Furthermore, MFI analysis also revealed a decreased expression of FoxP3 in the Treg subpopulation of the *Tusc2* KO males (*p =* 0.0089) and females (*p =* 0.0089) (Figure 6C) suggesting suppression of anti-inflammatory mechanisms in the *Tusc2* KO brain.

We also analyzed NK and NKT cell subsets. NK cells can regulate the adaptive immune responses and directly kill infected cells through the release of perforin and granzymes [57]. NKT cells also play diverse and essential roles in rapid response to infection, regulating immune responses, and linking innate and adaptive immunity [58].

Flow analysis revealed a significant decrease in NK cell population (CD11b^+^/CD3^−^/CD49^+^/Ly49G^+^) in female *Tusc2* KO brains (*p* < 0.0001), while no difference was found between male *Tusc2* KO and WT (Figure 7A,B); also, significant differences were observed between males and females of *Tusc2* KO (*p* < 0.0001) and WT mice (*p =* 0.0004). Analysis of NKT cells (CD11b^+^/CD3^+^/CD49^+^/Ly49G^+^) revealed a significant decrease in the overall proportion of these cells in *Tusc2* KO brains in both males (*p =* 0.0118) and females (*p* < 0.0001) (Figure 7C,D).

Lastly, we evaluated the population of B cells which are involved in recognizing and eliminating pathogens, formation of immunological memory, and mounting effective immune responses [59]. We detected a significant decrease in the number of activated B cells (CD19^+^/CD25^+^) in both male (*p* = 0.0343) and female *Tusc2* KO mice (*p* < 0.0001), as compared to their WT counterparts (Figure 7E,F). Overall, our results suggest that *Tusc2* deficiency disrupts immune populations in the brain, which could cause neuroinflammation.

### 2.3. Tusc2 Is Essential for the Homeostasis of CNS Proteins Regulating Intracellular Ca^2+^ Dynamics and Synaptic Plasticity

In the next step, we analyzed the expression of the key proteins involved in synaptic plasticity, energy metabolism, cytoskeleton structure, calcium signaling, and other brain functions that, when dysregulated, could lead to cognitive impairment. (Figure 8A). Glial fibrillary acidic protein (GFAP) plays a critical role in CNS glial cells, helping to maintain cell structure and support nearby neurons and BBB integrity [60]. The Western blot (WB) analysis revealed that GFAP protein was significantly decreased in the hippocampi of both male and female *Tusc2* KO mice (*p* = 0.0038) (Figure 8B). Calcium/calmodulin protein kinase II (CaMKII) involved in synaptic plasticity, calcium signaling, dendritic spine morphogenesis [61,62,63], etc., was significantly upregulated in both male and female *Tusc2* KO vs. WT mice (*p* = 0.0079) (Figure 8C). Another calcium-binding protein critical for neuronal health, Calbindin, [64] was also found to be significantly upregulated in male *Tusc2* KO hippocampi, but we did not see differences in Calbindin between female *Tusc2* KO and WT mice at this age (Figure 8D).

Chronic activation of the mTOR pathway in the brain is linked to protein aggregation, impaired autophagy, oxidative stress, neuroinflammation, and excitotoxicity. We used pS6/totalS6 protein ratio as an indicator of mTOR pathway activation and found it was significantly increased in female *Tusc2* KO hippocampi compared to WT females (*p* = 0.0330). Male *Tusc2* KO mice show a trend in upregulation in pS6/S6 that did not yet reach statistical significance at this young age (*p* = 0.0925) (Figure 8E). Our results suggest that Tusc2 plays a critical role in cognitive function by mediating key pathways involved in learning and memory formation, neuronal cellular structure, synaptic plasticity, proteostasis, and calcium-related processes. Moreover, the deficiency of Tusc2 promotes activation of the mTOR pathway, a critical pathology found in AD patients’ brains [65,66].

### 2.4. Human TUSC2 mRNA Levels Progressively and Significantly Decrease with Age across Blood and Brain Tissues

In order to investigate the potential association between the *Tusc2* KO model of premature aging and the aging process in humans, we analyzed the publicly available genotype-tissue expression (GTEx) database (https://gtexportal.org/) accessed on 15 March 2024. We followed changes in tissue-specific Tusc2 expression in distinct brain regions and whole blood using six different age groups. Notably, our linear regression analysis revealed a significant and progressive reduction in TUSC2 expression levels in blood immune cells and brain tissues with increasing age (whole blood: *p* < 0.00001; frontal cortex: *p* = 0.0003; hippocampus: *p =* 0.0002; amygdala: *p =* 0.0003) (Figure 9A). These results from the negative correlation between the expression of Tusc2 levels and age (Figure 9B) suggest that the changes related to the increase in the generation of inflammatory populations, as well as the decrease in spatial cognitive ability, could be closely related to the loss of Tusc2 expression during aging in humans, which might be fundamental in cognitive degeneration developing in early stages.

## 3. Discussion

Impaired cognitive function is the main symptom of Mild Cognitive Impairment (MCI) and AD. In our earlier study, we showed that 5-month-old female *Tusc2* KO mice have significant cognitive deficits when compared with WT females [18]. Here, we explored memory deficits in 4-month-old *Tusc2* KO mice of both sexes using the Y-maze test that measures short-term spatial working memory [67,68,69]. Four-month-old *Tusc2* KO mice also have deficits in working memory; however, males showed statistically significant differences while female mice showed only a trend (Figure 2). This observation led us to conclude that young male *Tusc2* KO mice age earlier than female *Tusc2* KO mice, although both sexes age earlier than WT mice. This conclusion is consistent with human studies that show that men biologically age earlier than women [70].

Chronic inflammation and neuroinflammation play a paramount role in the development and progression from MCI to the final stages of dementia [71]. Both innate and adaptive arms of immune response are involved in this process. Characterizing early changes in the brain innate and adaptive immune subsets that happen during early memory impairment (MCI stage) is essential. Several studies have noted the importance of the brain’s innate and adaptive immunity functions [72]. We specifically aimed to include minor immune subsets as less characterized subsets in human dementia. Neuroinflammation may initially start with the activation of CNS-resident immune cells, microglia, and astrocytes [71]. Microglia are the brain-resident macrophages involved in phagocytic clearance of cell debris and protein or other deposits in the brain [47,73], and, thus, play a role in all diseases of the CNS [74,75,76,77]. We did not observe Tusc2-dependent differences in the size or activation status of the microglial population (Figure 3 A,B). Since we analyzed microglia from young brains, we believe that pathological changes in the microglia of *Tusc2* KO mice could happen at a later age. 

Furthermore, through examining the proportions of both activated and resting microglia by evaluating the proportions of activated (CD45^bright^/CD11b^+^) and resting microglia (CD45^dim^/CD11b^+^) [52,53], our findings indicate that the percentage of the activated subpopulation in female mice from the *Tusc2* KO group was significantly higher compared to both the WT group and the *Tusc2* KO males (Figure 3C–E). These results suggest the possibility of a sex-specific effect of *Tusc2* KO on microglia activation despite maintaining equal proportions of microglia between the Wild-type and *Tusc2* KO groups. However, an analysis of additional subtypes of microglia related to neurodegeneration will be helpful in understanding if it plays a role in Tusc2-related memory impairment.

Astrocytes are the most abundant cell type of the CNS, outnumbering neurons in the human brain [78]. They play a key role in metabolic homeostasis, antioxidant defense, energy storage, mitochondria biogenesis, tissue repair, and synapse modulation by transferring mitochondria to neurons and supplying the building blocks of neurotransmitters [78]. Thus, mitochondrial and Ca^2+^ dysregulation caused by *Tusc2* loss in astrocytes could be detrimental to astrocyte functions and, thus, brain cognitive health. Astrocytes are crucial regulators of innate and adaptive immune responses in the injured CNS [58,79]. Interestingly, female *Tusc2* KO mice have a remarkable increase in astrocyte population compared to female WT mice (Figure 3F,G), suggesting pathological proliferation (activation) of astrocytes in female Tusc2 brains caused by a pro-inflammatory state of astroglia. When the brain is injured or diseased, astrocytes respond by proliferating and increasing in size [80]. This is called astrogliosis, and it is a common event in AD patients’ brains [34]. Our early studies showed that *Tucs2*-deficient T cells, peritoneal macrophages, and epithelial and fibroblast cells display chronic pro-inflammatory signatures [8,9,11,12,81,82]. Thus, astrogliosis in *Tusc2* KO brains is consistent with the pro-inflammatory phenotype of *Tusc2*-deficient cells. What was intriguing is that only the female brain showed astrogliosis at the age of 4 months, while male WT and *Tusc2* KO astrocyte populations were indistinguishable, suggesting that sex plays a significant role in the Tusc2-dependent immune responses of the brain.

The central nervous and peripheral immune systems have co-evolved; thus, crosstalk between immune pathways and neuronal circuits influences neurological diseases and behavioral responses [83]. The mechanisms responsible for the vicious, inflammatory loop in the brain that turns into chronic disease are not well understood. One possible mechanism is based on persistent inflammation coming from the periphery that can permanently change cognitive and behavioral states and lead to neurodegenerative disorders [84]. Molecules associated with adaptive immune responses, such as interleukin 4 (IL-4), interferon-γ (IFN-γ), and interleukin 17 (IL-17), have been associated with neurological behaviors and AD [85,86,87,88]. 

We observed multiple Tusc2- and sex-dependent changes in adaptive immune populations isolated from the brain. Thus, we found a dramatic decrease in Treg lymphocytes in male and female *Tusc2* KO mice (Figure 6A–C). Under normal physiological conditions, Treg cells have been shown to dampen the inflammatory responses of effector T cells, thus suggesting a neuroprotective role of these cells [89]. Moreover, ex vivo expansion of Treg cells with amplified immunomodulatory function suppressed neuroinflammation and alleviated AD pathology in vivo, thus directly implicating these cells in protection from neurodegeneration [90]. Our finding of severe reduction in Treg lymphocyte number suggests a neuroprotective role of Tusc 2 protein in the brain.

IFN-γ expression is controversial in the brain, with its activation of specific immune cells being neuroprotective or neurodegenerative [91]. It has been found that IFN-γ priming of microglia can induce proliferation and activation of these brain-resident macrophages, which can contribute to mechanisms that contribute to cognitive impairment and T cell infiltration early on in the neurodegenerative process [92]. We found a significant increase in IFN-γ-expressing Th_1_ cells in female *Tusc2* KO vs. female WT mice (Figure 4C–E). IFN-γ-expressing Th_1_ cells have been associated with memory function and hippocampal neurogenesis [93,94]. Conversely, Th_1_ cells have also been associated with the development of Parkinson’s disease in both experimental animal models and humans [89]. Thus, we suggest that increased expression of IFN-γ in Th_1_ cells in the female *Tusc2* KO brain, similar to that seen in patients with neurodegenerative diseases, is involved in the development of cognitive impairment.

Furthermore, we found a significant decrease in CD8^+^/CD25^+^ and CD8^+^/CD28^+^ T cells in both *Tusc2* KO males and females (Figure 5). However, female Tusc2 KO mice exhibit a trend of increased granzyme B expression in CD8^+^/CD28^+^ T cells. A recent study suggests that the infiltrated activated CD8 lymphocytes can trigger the pro-inflammatory response, directly contributing to neurodegeneration [95].

The exact role of NK cells in neurodegenerative disease is controversial, with these cells capable of ameliorating disease or exacerbating pathology [96,97,98]. In our study, we found that inhibitory mechanisms mediated by NK cells are disrupted in the *Tusc2* KO females based on the significantly decreased expression of Ly49G (Figure 7A,B). This receptor contributes directly to the education and self-tolerance of the NK cells [99]. Studies in mice and humans reported that NK cells from AD patients are more reactive, observing an increase in the production of IFN-γ and TNF-α, which is associated with a significant cognitive impairment. On the contrary, the depletion of these reactive populations led to improved cognitive function by reducing inflammation [100,101].

In addition, this decrease in Ly49G expression was observed in the CD11b^+^/CD3^+^/CD49^+^/Ly49G^+^ NKT cell population (Figure 7C,D). NKT cells are still being studied, and their role in specific organs, such as the brain, is not yet well understood. However, studies that have examined the role of other inhibitory receptors of the Ly49 family in NKT cells suggest that these receptors are crucial in upregulating mechanisms like the production of IL-10 while reducing the production of IFN-γ in these cells [102]. These findings suggest that the decreased populations of NK and NKT cells in the brain of *Tusc2* KO models may be crucial in regulating inflammation in the brain through Ly49G expression, which may be vital in preventing the development of a pro-inflammatory environment in the brain. Therefore, the malfunctioning of Ly49G could be a significant factor in the development of neuroinflammation. 

Our study also provided information on the significant differences in brain immune populations between males and females, which is critical for understanding the role of sex in immune responses and the development of pathologies of the brain. In WT mice, we found significant sex-dependent differences between males and females in the sizes of microglial (TMEM 119^+^/F4-80^+^) populations, CD4^+^/CD25^+^ T cells, CD8^+^/CD28^+^/GrzB^+^ T cells, and Ly49G^+^ NK cells, thus confirming profound sex-dependent differences in immune responses. In addition, we observed statistically significant differences between male and female *Tusc2* KO mice in CD8^+^/CD28^+^ T cells, CD3^+^/CD4^+^/Foxp3^+^ Treg cells and Ly49G^+^ NK cells. These findings suggest an intricate and complex dependence of the brain’s immune cell populations and responses to the constantly changing brain microenvironment due to sex and other complex factors. Thus, all studies performed on mice and humans should be sex-conscious.

Further analysis of Tusc2-dependent changes in critical neurodegeneration-linked proteins from hippocampal tissues showed a moderate but significant decrease (Figure 8B) in GFAP in both male and female *Tusc2* KO brains. GFAP is thought to help maintain astrocyte structural integrity and mechanical strength and aid cell movement and shape change [60]. Multiple CNS disorders are associated with improper GFAP expression, both up- and downregulated [103]. In fact, several possible mechanisms could explain the relationship between GFAP protein levels and Tusc2 loss. First, GFAP protein levels are known to be linked to changes in Ca^2+^ through calcium-dependent binding, calcium-/calmodulin-dependent phosphorylation, and calcium-dependent proteolysis [104]. As Tusc2 regulates Ca^2+^ homeostasis, its loss could result in GFAP dysregulation [9,81,82]. Second, multiple studies suggest that GFAP levels could be upregulated or downregulated depending on the context of the inflammatory environment [105]. Hence, TNF-α could increase astrocyte differentiation and proliferation while decreasing GFAP expression at both the messenger and protein levels due to the inhibition of STAT3 function [106,107]. Based on our results, we hypothesize that the absence of Tusc2 in the brain could lead to an increased Th_1_ response through IFN-γ and TNF-α and a decrease in anti-inflammatory subpopulations, which could result in an increase in the number of astrocytes with a decreased expression of GFAP. These disruptions could lead to a subsequent slowdown in the movement of astrocytes, which is critical for the functioning of neurons, synapses, and microglia in the *Tusc2* KO brain. However, further experiments are required to establish a clear cause-and-effect relationship between the loss of Tusc2, decreased GFAP expression, and astrocyte movement.

Calcium signaling in neurons connects membrane excitability with the biological function of the cell. The “calcium hypothesis” states that deregulation of calcium signaling is one of the early-stage and key processes in the pathogenesis of neurodegenerative diseases [108]. We showed in several studies that Tusc2 is involved in the regulation of Ca^2+^ fluxes to/from mitochondria [9,11,81,82]. Here, we checked for changes in proteins intimately involved in Ca^2+^ signaling (Figure 8C,D): CaMKII (calcium–calmodulin (CaM)-dependent protein kinase II) and Calbindin. CaMKII was significantly upregulated in both *Tusc2* KO females and males. Calbindin showed significant upregulation only in Tusc2 males at this early age. Further research should focus on identifying the entire network of Tusc2-dependent Ca^2+^-associated proteins to understand the scope of Tusc2 involvement in early events linked to neurodegeneration.

The mTOR pathway is implicated in many aging-related diseases and contributes to many age-related diseases, including neurodegeneration [109]. The phosphorylation of the S6 ribosomal protein (pS6) is downstream of mTOR activation and is a reliable marker of mTOR activity [110]. Based on an increased pS6 index (pS6/total S6), the mTOR pathway is significantly activated in the hippocampi of *Tusc2* KO females and showed a trend of activation in Tusc2 males (Figure 8E).

Thus, *Tusc2* loss caused a disbalance of pro-inflammatory and anti-inflammatory immune subsets and dyshomeostasis of Ca^2+^-dependent pathways critical for synaptic plasticity and memory formation, creating an unfavorable environment that may initiate aging-related neurodegeneration. 

### Conclusions

Our study demonstrated that the ubiquitous knock-out of *Tusc2*, a protein with a pivotal role in Ca^2+^ homeostasis, overall mitochondrial activities, and inflammation, is critically and mechanistically involved in early dysregulated events in the brain, resulting in cognitive and neuroimmune pathologies (Figure 10). Data also show progressively decreased expression of *Tusc2* in human tissues with aging, suggesting that the *Tusc2* KO model could, possibly, recreate early events in the brain leading to neurodegeneration. Further investigation into these early processes will lead to a better understanding of the course of development of neurodegenerative diseases and possibly lead to *Tusc2* or *Tusc2*-dependent processes being a therapeutic target for treating MCI/AD.

## 4. Materials and Methods

### 4.1. Mice 

We conducted our study using male and female WT and KO mice from a 129/sv background generated as described previously [13]. Animals were housed in an AAALAC International-accredited facility and in accordance with established guidelines and protocols approved by Meharry’s Institutional Animal Care and Use Committee. To maintain an optimal living environment, we housed 3–4 animals per cage in standard mouse cages with corn cob bedding. The vivarium was environmentally controlled with a temperature of 21 ± 1 °C, 30–70% humidity, and a 12 h light-dark cycle. The mice had access to food and water ad libitum. All animal protocols complied with the National Institutes of Health Guide for Care and Use of Laboratory Animals and were approved by the Institutional Animal Care and Use Committee of Meharry Medical College (animal protocol approval #16-07-582, dated 25 August 2020).

### 4.2. Behavioral Testing

The sequence of behavioral testing was novel object recognition (to measure object recognition memory), open-field activity (to measure exploratory activity), and the Y-maze test (to measure short-term spatial memory). The time between each test was 24–48 h. The following evaluation criteria were set to obtain accurate NOR, OFT, and Y-maze results: Test videos were recorded on the test day. Following testing, the videos were evaluated by two independent investigators who were blind to experimental conditions, i.e., genotype and sex. Findings from the two investigators were compared; if the findings of both were comparable, the average of the two results was taken. The light conditions were in accordance with AAALAC international standards, with light levels of 325 lux approximately 1 m (3.3 ft) above the floor.

### 4.3. Novel Object Recognition Test

This procedure was adapted from two research studies [111,112]. The NOR test involves three sessions: habituation, training, and testing. The testing apparatus was a classic open-field plastic container (i.e., white PVC plastic, 49 cm × 34 cm with walls 49 cm high). The container was covered from the outside with white, matte, non-reflective, heavy-duty paper to avoid reflections. During the habituation session, a mouse is placed in the arena without any objects and allowed to explore freely for 20 min. Afterward, the mouse is returned to its home cage. In the next session (training), the mouse is allowed to explore 2 identical objects. Finally, during the testing session, one of the training objects is replaced with a novel object. Because mice have an innate preference for novelty, if a mouse recognizes a familiar object, it will spend less time with the object and more of its time on the novel object [113]. The two objects used were a clear plastic bottle that was suitable for the size of the mice, and the other object consisted of multiple colored Lego blocks that were connected together. The Lego structure was also suitable for the size of the mice to encourage exploration. All results are recorded for further analysis. The following calculation was performed to determine the recognition object index (ROI)%: [(*percentage of exploration of the non-displaced familiar object during the training*) − (*percentage of exploration of the non-displaced familiar object during the test*)]/*percentage of exploration of the non-displaced familiar object during the training*. The recognition object index (ROI) measures the ability of an animal to recognize the same object at different time points. When the mouse remembers the familiar object (presented to the animal earlier), the exploration time of the object decreases. This task relies on different brain structures, mainly the hippocampus and perirhinal cortex.

### 4.4. Open Field/Locomotor Activity Test

The open-field test used specific measurements and zoning adopted from Sakamoto T. et al. 2019 [114]. The testing apparatus was a classic open-field plastic container (i.e., white PVC plastic, 49 cm × 34 cm, with walls 49 cm high). A 40% central zone was defined as the center area of the arena, while the peripheral area was defined as 60%. Each mouse was placed in the center of the arena, and their performance was recorded using a mounted camera recorder, Sony Digital HD Handycam (Sony Electronics Inc., San Diego, CA, USA). Their movement was recorded for 30 min after a prior acclimation phase of 30 min. Movement within the peripheral zone was counted if the mouse’s body was 75% or more in the zone. Completing the open-field test, we measured the time spent in the peripheral and central zones. The peripheral and central movements were defined as follows: peripheral = 4-paw movement in the peripheral outlined zone, with all 4 paws within the zone, excluding rearing behavior. Central movement = 4-paw movement in the central defined zone, with all 4 paws within the zone, excluding rearing behavior.

### 4.5. Y-Maze Test

This method was adapted from Maurice, T., 1996; Deacon and Raulins, 2006 [67,68]. The process involves placing mice individually into one arm of the maze and allowing them to explore it freely for 8 min. The Y-maze apparatus was made out of gray acrylic plastic. The floor was 8 cm in width, each arm was 36 cm in length, and the walls were 15.5 cm in height. The arms of the maze were interconnected at an angle of 120°. The number of entries into each arm is recorded, with the criteria that the mouse’s whole body (excluding its tail) must enter an arm to count as an entry. The mouse must make a correct combination of entries to be counted as a correct alteration [ABC, ACB, BCA, BAC, CAB, CBA]. We calculate the percentage of correct alterations using the following calculation:[(number of correct combinations)/(number of total entries−2)]×100

### 4.6. Western Blot Analysis

Hippocampal tissue was excised, flash-frozen, and stored in liquid nitrogen until processing. The tissue was homogenized and lysed in RIPA buffer, and the protein concentrations were determined using a BCA protein assay (Bio-Rad, Hercules, CA, USA). An equal concentration of proteins was loaded on SDS-PAGE pre-cast gels (Bio-Rad, Hercules, CA, USA) and separated via electrophoresis. The proteins were transferred to a nitrocellulose membrane using a Turbo Blot system (Bio-Rad, Inc.) and stained with Ponceau stain for protein load visualization. The membranes were blocked with 5% BSA for 1hr at room temperature, followed by incubation with primary antibodies in 5% BSA overnight at 4 °C. The membranes were washed in 1xTBST and incubated for 2 h with the corresponding secondary antibodies. The blots were then visualized using the ECL chemiluminescence substrate and imaged with the GE Amersham™ Imager 600 (GE Amersham™ imager 600, Waltham, MA, USA). Band intensities were quantified using ImageJ software, version 1.54g (NIH, Bethesda, MD, USA) and normalized to Ponceau-stained proteins/lane bands.

The following primary antibodies were used: ThermoFisher Scientific (Waltham, MA, USA): Anti-mouse GFAP monoclonal (1:500, Clone GA5, Cat. #14-9892-82), Anti-mouse Calbindin monoclonal (1:1000, Cat. # 702411); Cell Signaling, Inc. (Danvers, MA, USA): Anti-mouse S6 ribosomal protein (1:1000, Clone 54D2, Cat. # 2211S), Anti-mouse Phospho-S6 ribosomal protein monoclonal (1:500, Clone Ser235/236 Cat. # 2371S); Santa-Cruz Biotechnology (Dallas, TX, USA): Anti-mouse CaMKII monoclonal (1:1000, Clone A-1, Cat. # SC-13141). Secondary antibodies: Goat Anti-rabbit IgG (1:6000, Cat. # 1706515) or Goat Anti-mouse IgG (1:6000, Cat. # 1706516).

### 4.7. Isolation of Brain Immune Cells

Mice were deeply anesthetized with isoflurane, followed by sacrifice via cervical dislocation. Whole brains were excised and placed in individual Petri dishes on ice containing HBSS/5%FBS for processing. The whole brains were thoroughly chopped, followed by adding 5 mL of mild-enzymatic digestion solution according to the manufacturer’s protocol (Cat. # 07473, Stem Cell Technologies, Vancouver, BC, Canada) and digestion for 20 min in a shaking cell culture incubator. Cold HBSS/5%FBS was added to the digestion solution to stop the enzymatic reaction. The digested tissue was homogenized with a Pasteur pipette and filtered through a 70 µm cell strainer. The cells were washed with cold HBSS/5%FBS and centrifuged for 5 min at 1500 rpm. The supernatant was removed and replaced with 5 mL of 30% Percoll, followed by centrifugation at 2300 rpm for 30 min at 18 °C. After centrifugation, the cells were washed three times and counted using Trypan blue stain with an automated cell counter, Countess (ThermoFisher Scientific).

### 4.8. Cell Staining and Flow Cytometry Analysis

To analyze the resident and infiltrating brain immune cells, we designed the staining antibody cocktails specific to the following types of cells: microglia, astrocytes, cytotoxic CD8 T cells, CD4 T helper (Th1), regulatory T cells (Treg), Natural killer (NK) and Natural killer T cells (NKT). For surface staining, we used the following: Anti-mouse Tmem119-PE (1:50, Cat. # 12-6119-82), Anti-mouse ACSA-2-APC (1:00, Cat. # 130-116-142), Anti-mouse CD45-PE/Cyanine (1:400, Cat. # 103114), Anti-mouse CD11b-BV650 (1:400, Cat. # 101239), Anti-mouse F4/80-AF594 (1:50, Cat. # 123140), Anti-mouse CD8-BV605 (1:400, Cat. # 100743), Anti-mouse CD4-APC (1:100, Cat. # 100412), Anti-mouse CD25-FITC (1:300, Cat. # 102006), Anti-mouse CD19-BV650 (1:400, Cat. # 115541), Anti-mouse CD3-BV421 (1:400, Cat. # 100227), Anti-mouse CD3-BV711 (1:400, Cat. # 100241), Anti-mouse Ly-49G2- FITC (1:100, Cat. # 11-5781-82), Anti-mouse CD25-APC/Cyanine7 (1:50, Cat. # 102026), Anti-mouse CD28-PE (1:50, Cat. # 102106), and Anti-mouse CD49b-PE (1:400, Cat. # 12-5971-82). For intracellular staining, we used the following: Anti-mouse IFN-γ-FITC (1:50, Cat. # 505806), Anti-mouse TNF-α-PerCP/Cyanine5.5 (1:50, Cat. # 506322), Anti-mouse IFN-γ-BV650 (1:50, Cat. # 505831), Anti-mouse Granzyme B-FITC (1:50, Cat. # 11-8898-82), and Anti-mouse FOXP3- PE (1:100, Cat. # 126404). All antibodies were purchased from Biolegend^®^ (San Diego, CA, USA), ThermoFisher Scientific, or Miltenyi Biotec, USA (San Diego, CA, USA). Single-color control staining and compensation beads were used as controls for gating and compensation. Live cells were gated based on positivity for Zombie Aqua viability stain (1:600, Biolegend^®^, cat. # 423101), followed by gating for specific cells of interest. All the experiments were performed using the Cytex Amnis CellStream benchtop flow cytometer (Cytek^®^ Biosciences, Fremont, CA, USA) and analyzed using FlowJo software v.10.9 (Treestar Inc., Woodburn, OR, USA).

### 4.9. Statistical Analysis

All the values were expressed as mean ± standard error of the mean (SEM). Statistical analysis was performed using GraphPad Prism v10.0.3. In order to observe the normality and homogeneity distribution, the data were subjected to the D’Agostino normality test and subsequently to Student’ *t*-tests (unpaired, two-tailed) to assess the statistical significance of differences between the two groups. Multiple groups were compared using two-way ANOVA, followed by Tukey’s multiple comparisons test or Bonferroni’s multiple comparisons test for significance of difference, where appropriate. The significance level was set at the minimum to *p* ≤ 0.05 for all statistical analyses.

## Figures and Tables

**Figure 1 ijms-25-07406-f001:**
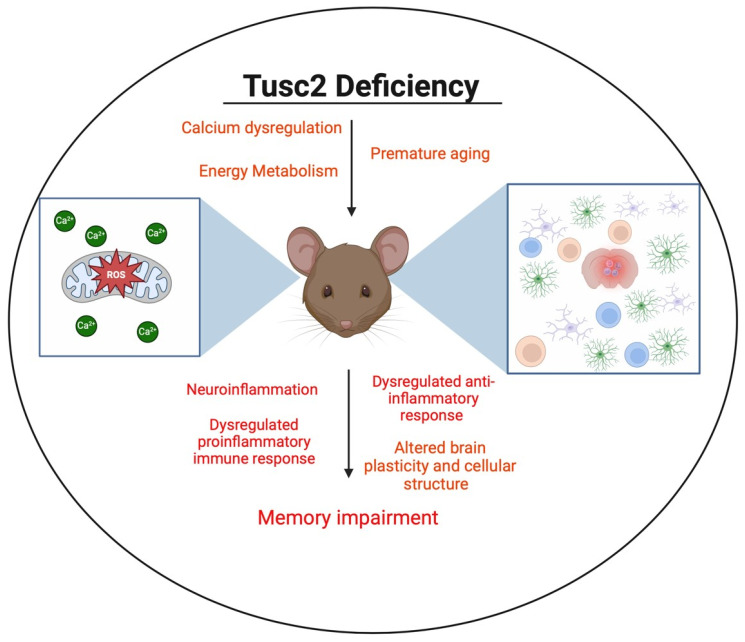
Illustration representing the impact of Tusc2 deficiency on the central nervous system.

**Figure 2 ijms-25-07406-f002:**
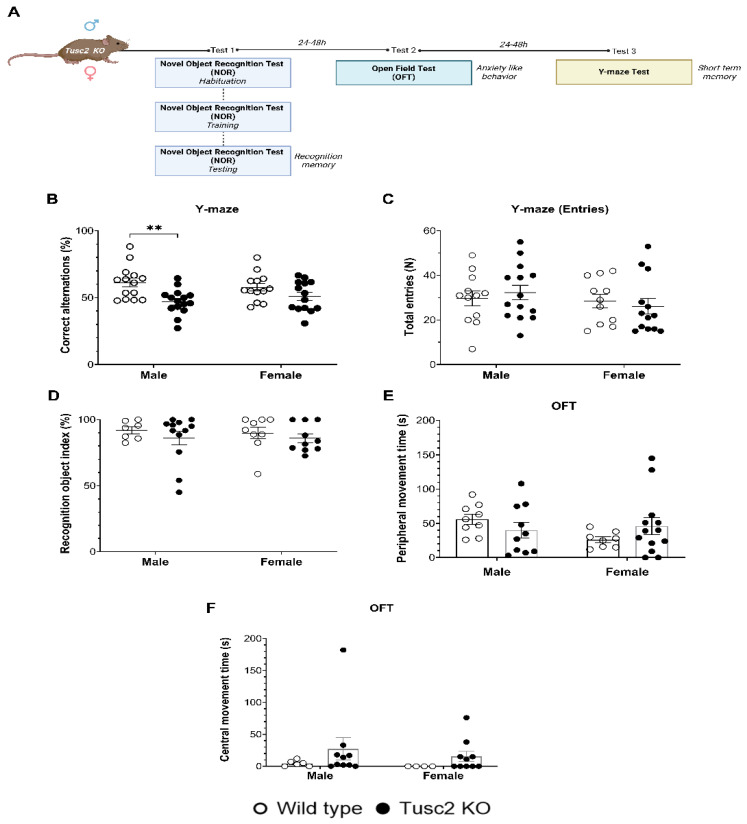
Tusc2 deficiency causes impairment in short-term spatial memory. (**A**) A graphic representation of the cognitive evaluation in the Tusc2 KO model where the recognition memory, anxiety-like behavior, and short-term memory were evaluated. (**B**) Y-maze test. Correct alternations (%) within an 8-min period is shown; Tusc2 KO males have a significant decrease in % correct alternations (F_(1, 52)_ = 12.74, *p* = 0.0055), as compared to WT males. Female WT and KO showed no significant difference in the Y-maze test. Interaction df = 1, F (DFn, DFd) = F_(1, 52)_ = 1.596, *p* value = *p* = 0.2122. Sex df = 1, F (DFn, DFd) = F_(1, 52)_ = 0.006853, *p* value = *p* = 0.9343. Gene df = 1, F (DFn, DFd) = F_(1, 52)_ = 12.74, *p* value = *p* = 0.0008. (**C**) The total number of entries within the 8 min was recorded; no significant difference was seen in performance between groups. (**D**) Novel object recognition (NOR) tests. NOR: ROI index during a 3-min period. Tusc2 male and female KO and WT mice performed consistently in the NOR task; no significant difference was seen across groups. Interaction df = 1, F (DFn, DFd) = F_(1, 34)_ = 0.05171, *p* value = *p* = 0.8215. Sex df = 1, F (DFn, DFd) = F_(1, 34)_ = 0.05562, *p* value = *p* = 0.8150. Gene df = 1, F (DFn, DFd) = F_(1, 34)_ = 1.173, *p* value = *p* = 0.2863. (**E**) Open-field (OF) test, 30 min of active movement in a peripheral zone: Interaction df = 1, F (DFn, DFd) = F_(1, 36)_ = 2.760, *p* value = *p* = 0.1053. Sex df = 1, F (DFn, DFd) = F_(1, 36)_ = 1.226, *p* value = *p* = 0.2755. Gene df = 1, F (DFn, DFd) = F_(1, 36)_ = 0.03526, *p* value = *p* = 0.8521. (**F**) Open-field (OF) test, 30 min of active movement in a central zone: Interaction df = 1, F (DFn, DFd) = F_(1, 26)_ = 0.64, *p* value = *p* = 0.8022. Sex df = 1, F (DFn, DFd) = F (1, 26) = 0.3293, *p* value = *p* = 0.5710. Gene df = 1, F (DFn, DFd) = F (1, 26) = 1.844, *p* value = *p* = 0.1861. All data underwent Tukey’s multiple comparisons test followed by two-way ANOVA. ** *p* ≤ 0.01. *n* = 7–16 mice per group.

**Figure 3 ijms-25-07406-f003:**
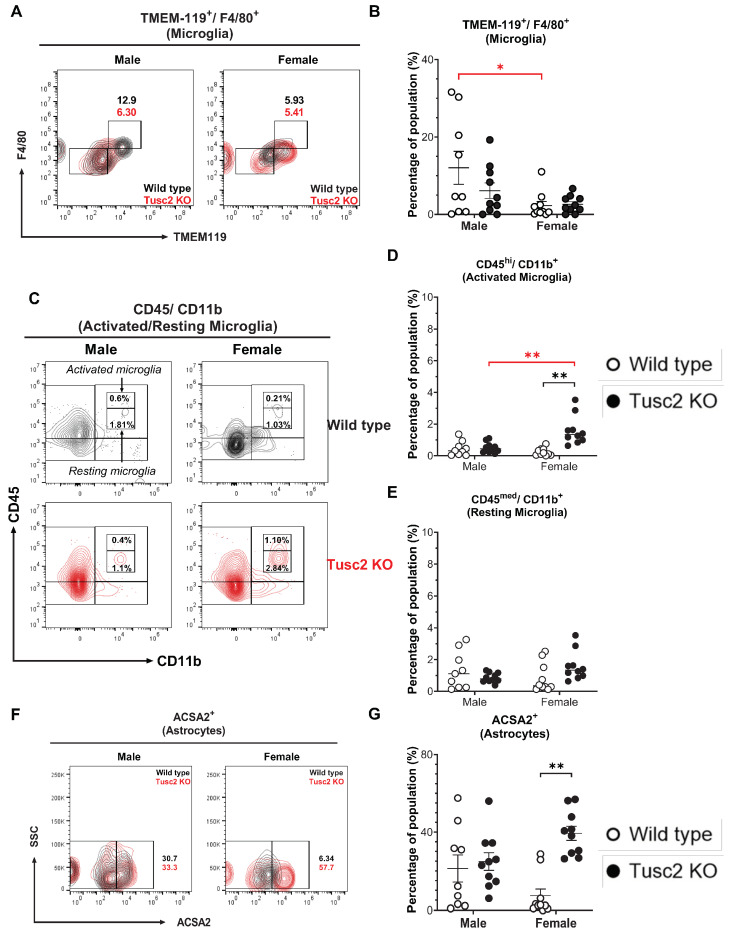
The loss of *Tusc2* causes increases in brain-resident immune cells. (**A**) Representative contour dot plot graph comparing the differences of Microglia population (TMEM-119^+^/F4/80^+^) between the Wild type (black) and the *Tusc2* KO (red) models in males and females. (**B**) shows the percentage of microglial cells within the CNS. *Tusc2* WT and KO male proportions are comparable, with no significant difference (*p* = 0.0356). *Tusc2* WT and KO female proportions are comparable, with no significant difference. (**C**) Representative contour dot plot graph showing the differences of activated and resting population (CD45^bright^/CD11b^+^ and CD45^dim^/CD11b^+^, respectively) between the Wild type (black) and the *Tusc2* KO (red) model in males and females. (**D**) shows the differences in the percentages of activated populations between the WT and KO groups (*p* = 0.0006) (**E**) and resting microglial cells with no significant changes. (**F**) A representative contour dot plot graph comparing the differences in the Astrocytes population (ACSA2^+^) between the Wild type (black) and the *Tusc2* KO (red) models in males and females. *Tusc2* male WT and KO percent astrocytes are comparable. (**G***) Tusc2* KO female mice exhibited a significant increase in the percentage of astrocytes in the CNS (*p* = 0.0002) compared to their WT counterparts. Two-way ANOVA followed by Bonferroni’s multiple comparisons test. * *p* ≤ 0.05, ** *p* ≤ 0.01. *n* = 10 mice per group. For bar graphs, the black asterisk denotes comparisons between the WT and KO models, while the red asterisk represents the comparison between the sexes. “Percentage of population” indicates the proportion of all events in the population gating shown at the top of the graphic.

**Figure 4 ijms-25-07406-f004:**
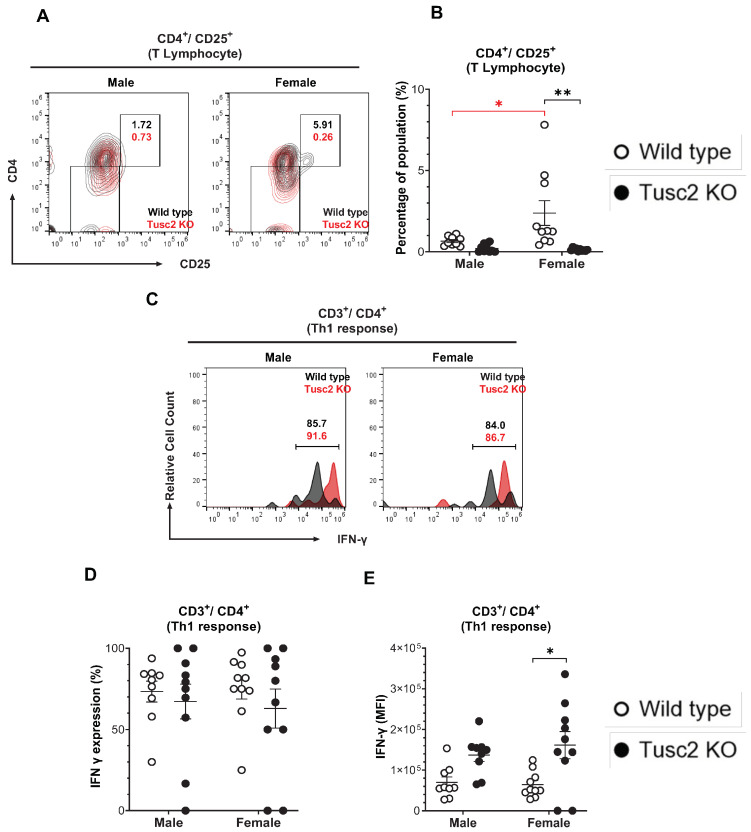
*Tusc2* deficiency causes alterations in CD4^+^ T and Th_1_ cells in the brain. (**A**) Representative contour dot plot graph comparing the differences of T lymphocyte population (CD4^+^/CD25^+^) between the Wild type (black) and the *Tusc2* KO (red) models in males and females. (**B**) CD4 T cell proportions in the CNS were significantly decreased in KO females (*p* = <0.0001); no significant difference was observed in males. (**C**) Representative histogram showing the difference in IFN-γ expression between CD4^+^/CD25^+^ T lymphocytes (Th_1_ response) of Wild type (*black*) and *Tusc2* KO (*red*) male and female models. (**D**) CD4 T cell subset Th_1_ cells were evaluated. (**E**) MFI values of IFN-γ cytokine-producing CD4 T cells were significantly decreased in *Tusc2* KO female mice (*p* = 0.0068); male KO mice showed no significant difference (*p* = 0.2136) in proportion as compared to their WT counterparts. Two-way ANOVA followed by Bonferroni’s multiple comparisons test. * *p* ≤ 0.05, ** *p* ≤ 0.01. *n* = 10 mice per group. For bar graphs, the black asterisk denotes comparisons between the WT and KO models, while the red asterisk represents the comparison between the sexes. “Percentage of population” indicates the proportion of all events in the population gating shown at the top of the graphic.

**Figure 5 ijms-25-07406-f005:**
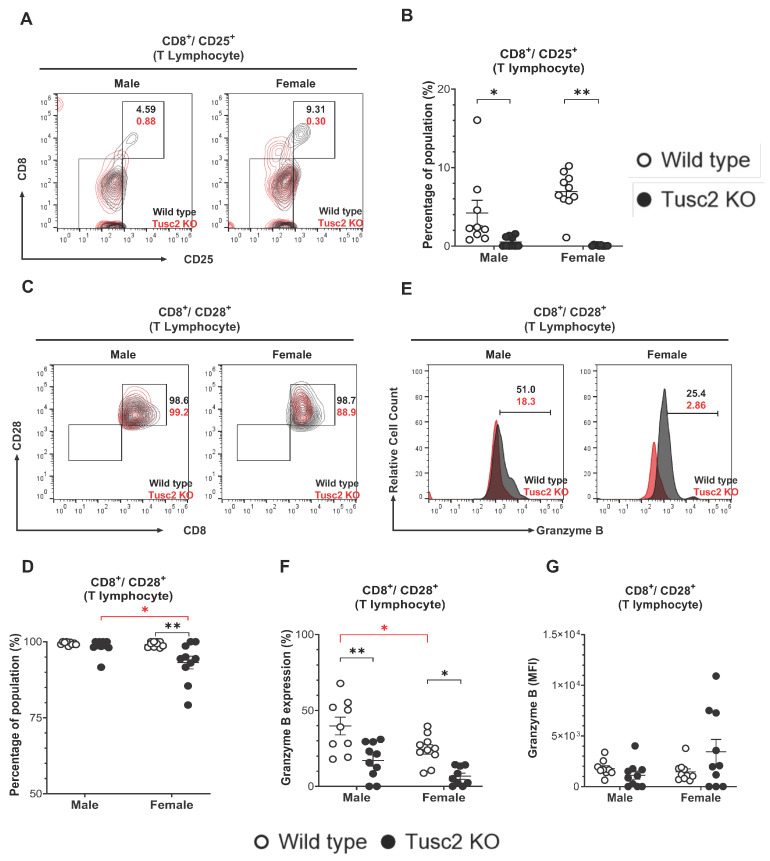
Loss of *Tusc2* affects the frequency of CD8^+^ T cells in the CNS. (**A**) Representative contour dot plot graph comparing the differences of cytotoxic T lymphocyte population (CD8^+^/CD25^+^) between the Wild type (black) and the *Tusc2* KO (red) models in males and females. (**B**) CD8^+^ T cells were evaluated; *Tusc2* was significantly decreased in KO males (*p* = 0.0006) and KO females (*p* = <0.0001). (**C**) Representative contour dot plot graph comparing the differences of T effector cytotoxic lymphocyte population (CD8^+^/CD28^+^) between the Wild type (black) and the Tusc2 KO (red) model in males and females. (**D**) The proportion of CD28^+^ cytotoxic CD8^+^ T cells was evaluated and was found to be significantly decreased in female *Tusc2*-KO mice. (**E**) Representative histogram showing the difference in granzyme B expression between CD8^+^/CD28^+^ effector cytotoxic T lymphocytes of Wild type (black) and *Tusc2* KO (red) male and female models. (**F**) Granzyme-B-expressing CD8^+^ T positive cells were shown to be significantly decreased in *Tusc2* KO males (*p* = <0.0001) and KO females (*p* = 0.0003). (**G**) MFI of granzyme-B-expressing CD8^+^ T cells did not show a statistical difference in expression. Two-way ANOVA followed by Bonferroni’s multiple comparisons test. * *p* ≤ 0.05, ** *p* ≤ 0.01. *n* = 10 mice per group. For bar graphs, the black asterisk denotes comparisons between the WT and KO models, while the red asterisk represents the comparison between the sexes. “Percentage of population” indicates the proportion over all events in the population gating shown at the top of the graphic.

**Figure 6 ijms-25-07406-f006:**
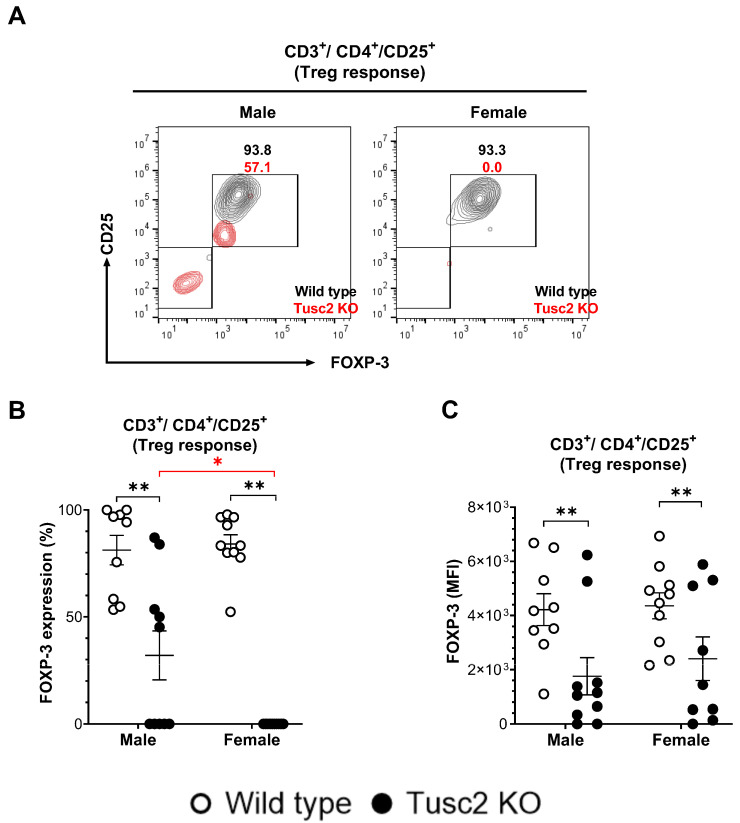
Lack of *Tusc2* disrupts the inhibitory mechanisms of immune cells responsible for regulating inflammation. (**A**). Representative contour dot plot graph showing the difference in FOXP-3 expression between CD3^+^/CD4^+^/CD25^+^ T lymphocytes (Treg response) of Wild type (black) and *Tusc2* KO (red) male and female models. (**B**,**C**) Flow analysis revealed a decrease in the overall proportion and MFI value of Treg cells for KO males (*p* = 0.0001) and KO females (*p* = <0.0001). Two-way ANOVA followed by Bonferroni’s multiple comparisons test. * *p* ≤ 0.05, ** *p* ≤ 0.01. *n* = 10 mice per group. For bar graphs, the black asterisk denotes comparisons between the WT and KO models, while the red asterisk represents the comparison between the sexes.

**Figure 7 ijms-25-07406-f007:**
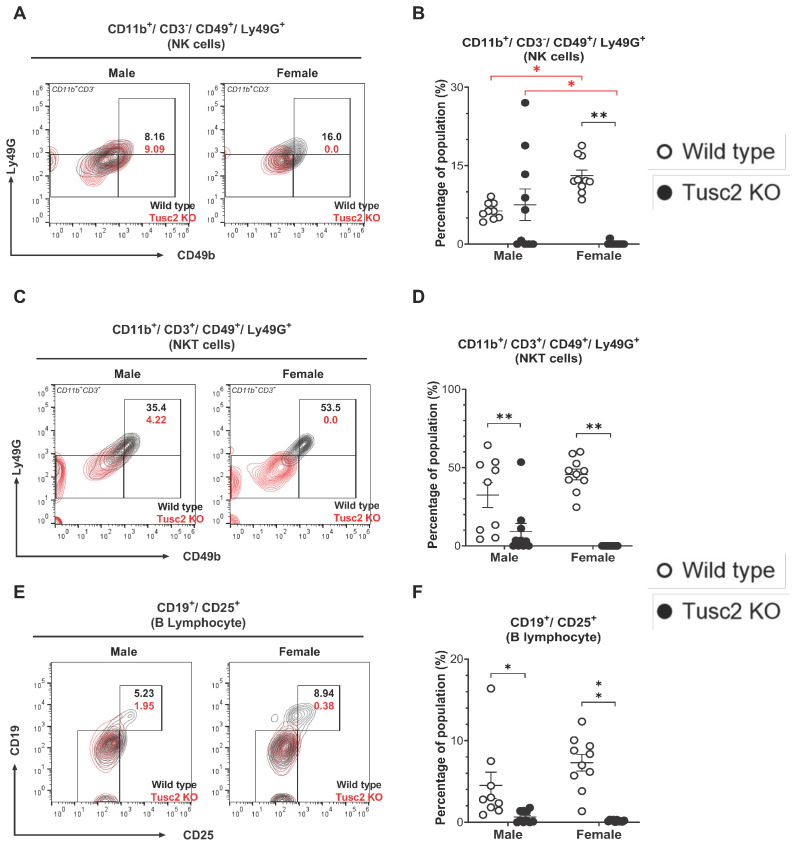
*Tusc2*-KO disrupts the frequency of NK and NKT cells in the CNS. (**A**) Representative contour dot plot graph comparing the differences in Ly49G^+^ NK cell population (CD11b^+^/CD3^−^/CD49^+^) between the Wild type (black) and the *Tusc2* KO (red) males and females. (**B**) *Tusc2* female KO mice showed a decrease in the proportion of NK cells (*p* = <0.0001) compared to their WT counterparts; KO males did not show a significant difference compared to WT males. (**C**) Representative contour dot plot graph comparing the differences in Ly49G^+^ NKT cell population (CD11b^+^/CD3^+^/CD49^+^) between the Wild type (black) and the *Tusc2* KO (red) model in males and females. (**D**) NKT cell expression was significantly decreased in *Tusc2* KO males (*p* = 0.0118) and in *Tusc2* KO females (*p* = <0.0001). (**E**) Representative contour dot plot graph comparing the differences of B lymphocyte population (CD19^+^/CD25^+^) between the Wild type (black) and the *Tusc2* KO (red) model in males and females. (**F**) The proportion of B cells was also evaluated. *Tusc2* KO males had a significant decrease in B cells (*p* = 0.0343), and *Tusc2* KO females showed a decrease in B cell proportions as well (*p* = <0.0001). Two-way ANOVA followed by Bonferroni’s multiple comparisons test. * *p* ≤ 0.05, ** *p* ≤ 0.01. *n* = 10 mice per group. For bar graphs, the black asterisk denotes comparisons between the WT and KO models, while the red asterisk represents the comparison between the sexes. “Percentage of population” indicates the proportion of all events in the population gating shown at the top of the graph.

**Figure 8 ijms-25-07406-f008:**
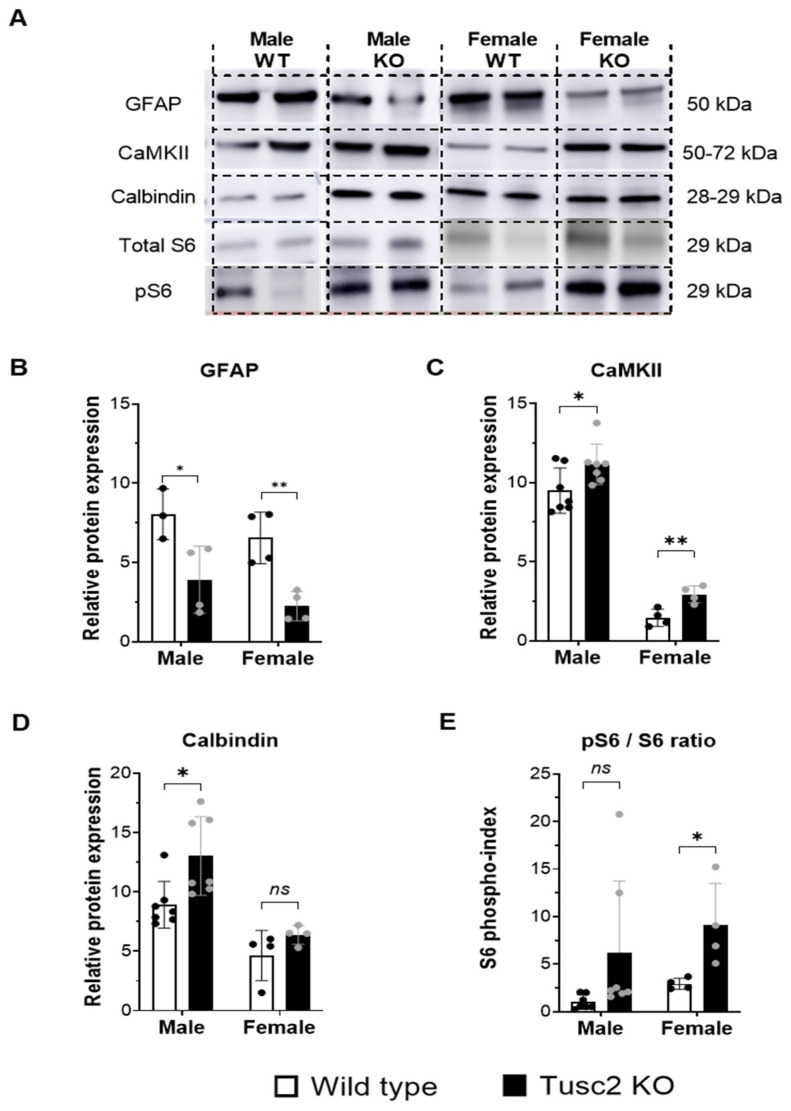
Western blot analysis of age-related proteins in the hippocampi of *Tusc2* wild type and KO mice. (**A**) Western blotting graphic showing the evaluation of the proteins involved in brain integrity, synaptic transmission, and CNS homeostasis. (**B**) Shows the relative expression of GFAP in WT and KO female and male *Tusc2* mice; KO males showed a significant decrease in GFAP (*p* = 0.0378, *n* = 3), and females showed a significant decrease as well (*p* = 0.0038, *n* = 4). (**C**) CaMKII levels in *Tusc2* KO males and female mice were significantly increased (*p* = 0.0336, *n* = 7) and (*p* = 0.0079, *n* = 4), respectively. (**D**) Levels of Calbindin were measured. *Tusc2* KO males showed a significant increase in expression (*p* = 0.0055, *n* = 7), while females showed no significant difference (*p* = 0.1721, *n* = 4). (**E**) Phosphorylated S6 was significantly increased in female KO mice (*p* = 0.0330, *n* = 4), and a trend of increase was seen in males (*p* = 0.0925, *n* = 7). Ponceau staining was used as a loading control All data were expressed as the mean ± SEM. Two-tailed unpaired *t*-test. * *p* ≤ 0.05, ** *p* ≤ 0.01. *n* = 3–4 mice per group. ns = not significant.

**Figure 9 ijms-25-07406-f009:**
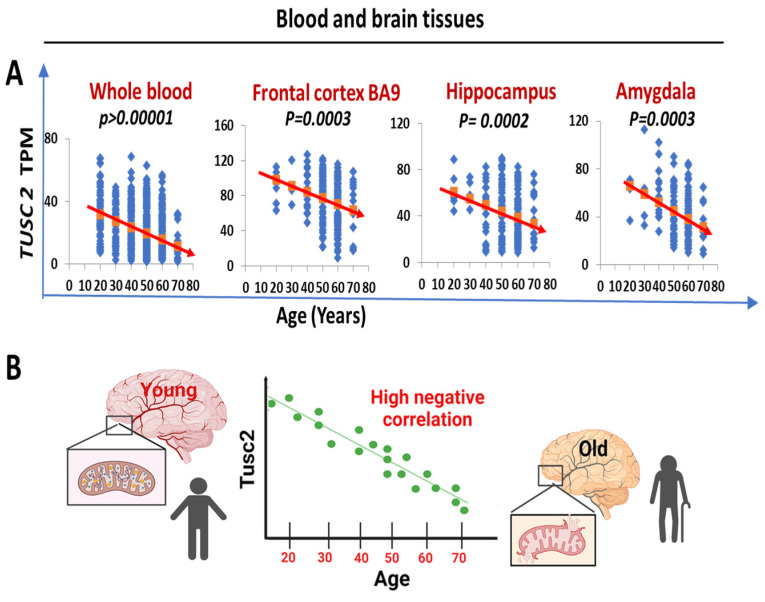
*TUSC2* mRNA levels are decreased with age in the human blood and brain tissues. (**A**) A linear regression analysis shows progressive and significant age-dependent downregulation of the TUSC2 levels (TPM). Blue dots represent the TUSC2 levels in a single specimen, red arrows show statistically significant downward trends in *TUSC2* expression in the groups of individuals ranging from 20 to 80 years old. The data were obtained from the Genotype-Tissue Expression (GTEx) database. All data are represented as the mean ± SD. *n* = 408 (68 samples per group). (**B**) Negative correlation observed between the age and *TUSC2* expression levels in humans (the higher the age, the lower *TUSC2* expression).

**Figure 10 ijms-25-07406-f010:**
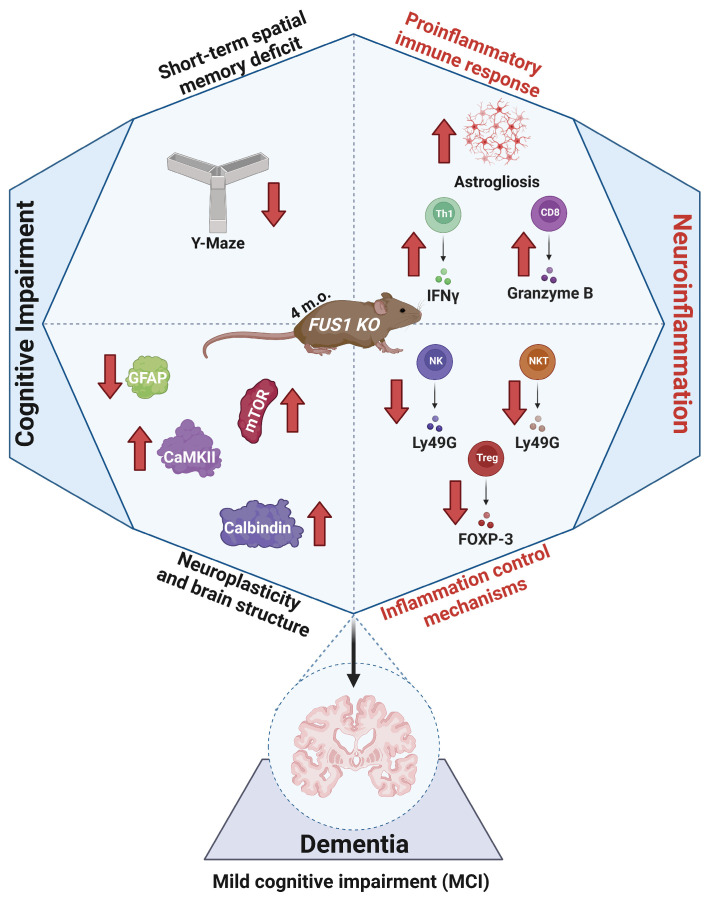
Graphical representation of key findings. The *Tusc2* model displays changes associated with cognitive impairment caused by neuroinflammatory factors. Additionally, *Tusc2* KO causes disturbances in the inhibitory mechanisms responsible for upholding the cellular microenvironment in homeostasis, leading to changes in crucial proteins responsible for maintaining the brain’s structure and neuroplasticity. These changes could potentially influence the development of cognitive impairment. Red arrows denote the direction of change; upward arrows indicate an increase while downward arrows indicate a decrease in the indicated molecules.

## Data Availability

The authors will be pleased to provide access to the raw data that support the conclusions of this article without any hesitation.

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
