# Peer review of "Loss of Mitochondrial Tusc2/Fus1 Triggers a Brain Pro-Inflammatory Microenvironment and Early Spatial Memory Impairment"

_ijms, 2024, doi:10.3390/ijms25137406_

Round 1
Reviewer 1 Report
Comments and Suggestions for Authors
The paper by Farris and colleagues aims to characterize Tusc2- age- and sex-dependent immune changes in the brain and “Alzheimers-specific” molecular changes in
the hippocampus, a part of the brain involved in short-term memory consolidation and
Storage. The paper has findings that will be of interest to the AD field. However, the behavioral tasks do not appear to be ran as rigorously as required to make the claims presented in the paper. This isn’t cause for rejection of the paper, but the statements in the conclusion should be toned down.
Major comments:
-
I don’t believe that any of the behavioral tasks performed in the study are accurately assessing spatial memory. For example, in the Y-Maze task, the authors are assessing spontaneous alterations which past studies have shown may not entirely be hippocampal dependent. Can I blind mice perform this type of task correctly? If anything, this may be a prefrontal cortex dependent but more care should go into the conclusions derived from the test. The authors state, “significant short-term spatial memory deficit in Y-maze tests based on lower percentage of correct alternation as compared to their WT male counterparts”. However, I believe this is more of a working memory task than a short-term spatial memory task. Additionally, the authors provide no data to show whether Tusc KO mice have a tendency towards repetitive or preservative behavior (or lack thereof) which impacts the interpretation of the test. Last, the authors should provide some data on anxiety-related novelty avoidance which could also impede the interpretation of the behaviors.
-
The authors found no differences between groups or sexes in other behavioral tasks, which is surprising because of the wealth of data that has found sex-differences in spatial long-term and working memory tasks. Can the authors speak more to this?
-
The NOR test should be ran more carefully in the future. There are multiple studies showing the sensitivity of this test, and one thing that sticks out to this reviewer is the objects used in the test. A clear water bottle and a colorful lego. Rodents enjoy novelty and so the inherent differences between these two objects could have played a role in why no differences were observed between groups. One suggestion would be to counterbalance the objects used in the task, and another is to find another object besides a clear water bottle that is more “exciting.” To overcome this, the authors should provide the amount of time the mice spent with the objects in the training trials.
-
Do mice have a gender? I think since this is referring to mice, “Gender” can be changed to “sex” throughout the paper.
Author Response
Please see the attached comments for the reviewer 1.

Reviewer 2 Report
Comments and Suggestions for Authors
The research article by Tonie Farris et al. presents a valuable contribution to the understanding of proinflammatory microenvironment and immune changes leading to the memory impairment in mitochondrial Tusc2/Fus1 knocked out mice model. The strengths lie in the clear research question, robust methodology, and significant findings. The introduction is reasonable well written and provides a good overview leading to clear problem statement. The abstract is informative, the results are well supported with the presented data and figures. The discussion is insightful and hence the manuscript is suitable for publication with the inclusion of following minor points for improvement, if possible.
- Size of the diagrams in the figure 2, 3, 4 5 and 6 should be improved, specially the contour maps and figures included in the supporting information.
- The numbers presented in figures should be incorporated in discussing the results leading up to the interpretation of the results.
- The authors could comment on the mechanistic reasons behind the significant differences in the results of male and female’s mice for few of the parameters included in this study.
- The authors can include a pictorial representations summarizing their results and key findings leading up to the imbalance of proinflammatory and anti-inflammatory parameters and memory impairment.
- The authors can also include the pictorial representation of the problem statement taken up in this study. The figure can be included in the introduction section.
